# Professional Competencies and Job Satisfaction Among Physiotherapists: Validation and Psychometric Analysis of the Multidimensional Scale

**DOI:** 10.3390/healthcare13202595

**Published:** 2025-10-15

**Authors:** Emanuela Prendi, Enkeleda Gjini, Florian Spada, Blerina Duka, Rosario Caruso, Francesco Scerbo, Giovanni Gioiello, Federico Ruta, Ippolito Notarnicola

**Affiliations:** 1Department of Biomedical Sciences, Faculty of Medicine, University “Our Lady of the Good Counsel”, 1000 Tirana, Albania; e.prendi@unizkm.al (E.P.); e.gjini@unizkm.al (E.G.); f.spada@unizkm.al (F.S.); bleriduka@yahoo.it (B.D.); 2Health Professions Research and Development Unit, Department of Biomedical Sciences for Health, IRCCS Policlinico, 20097 San Donato, Italy; rosario.caruso@unimi.it; 3Department of Biomedicine and Prevention, University of Rome Tor Vergata, 00133 Rome, Italy; scerbofrancesco@gmail.com; 4Department of Medicine and Surgery, University of “Kore”, Piazza dell’Università, 94100 Enna, Italy; giovanni.gioiello@unikore.it; 5Health Agency BAT, General Direction, 76123 Andria, Italy; federicorutabat@gmail.com

**Keywords:** physiotherapy, professional skills, personal mastery, job satisfaction, factor analysis, logistic regression, organizational well-being

## Abstract

**Background/Objectives:** Professional competencies and personal mastery are key dimensions for the well-being of health professionals and the quality of care. In physiotherapy, where organizational complexity is common, job satisfaction depends on both clinical skills and resilience. While these aspects have been explored in nursing, evidence for physiotherapists is limited. This study aimed to (1) assess perceived competencies and personal mastery in Italian physiotherapists; (2) analyze their relationship with job satisfaction; and (3) examine the factorial structure of the Multidimensional Scale of Competences. **Methods:** A cross-sectional study was conducted with 481 physiotherapists working in various care settings. Data were collected using the 25-item Multidimensional Scale of Competences, the 7-item Personal Mastery Scale, and a single job satisfaction item, all on a 5-point Likert scale. Analyses included descriptive statistics, Pearson correlations, logistic regression, and exploratory factor analysis (Principal Component Analysis with five components). **Results:** Participants had a mean age of 31.1 years (SD = 8.3) and 7.3 years of professional experience (SD = 7.7); gender distribution was balanced. Most held a master’s (44.5%) or bachelor’s degree (36.8%). Job satisfaction was high, with 95% reporting moderate to very high satisfaction. Competencies showed a mean of 4.16 (SD = 0.95; α = 0.86), while Personal Mastery averaged 3.52 (SD = 1.29; α = 0.60). Competencies significantly predicted job satisfaction (OR = 8.37, *p* = 0.003), whereas Personal Mastery did not. Factor analysis identified five domains—technical–clinical, communicative, collaborative, ethical, and educational—explaining 50.3% of variance. **Conclusions:** Italian physiotherapists report high competencies and moderate personal mastery. Job satisfaction is strongly linked to competencies, highlighting their central role in professional well-being. Results support the importance of continuous professional development and organizational strategies that enhance competencies and resilience.

## 1. Introduction

The growing complexity of health systems has made the issue of the skills of health professionals increasingly central, since the quality of care depends not only on up-to-date technical knowledge, but also on an articulated set of relational, ethical and organizational skills [1]. In the case of physiotherapy, the evolution of health needs, the aging of the population and the increase in chronic conditions have progressively expanded the role of the physiotherapist, who today is not limited to providing rehabilitation treatments, but is involved in prevention, patient education and the promotion of healthy lifestyles [2]. This transformation requires assessment tools capable of grasping the multidimensional nature of skills from a perspective that integrates clinical, communicative, ethical and professional updating dimensions [3,4].

The literature has highlighted how the concept of professional competence does not simply coincide with the set of technical skills possessed, but represents the ability to integrate knowledge, practical skills, values and attitudes in real contexts, often characterized by uncertainty and complexity [5]. Competence, in this sense, is manifested through professional behavior, the ability to adapt to new situations, to make appropriate clinical decisions and to collaborate in interdisciplinary teams. In physiotherapy, these skills translate into the ability to assess and treat patients with diversified needs, establishing relationships of trust and effective communication, while maintaining a solid ethical and deontological orientation [6]. The availability of valid and reliable tools to measure these skills is essential to monitor the quality of training, guide the planning of continuous training and foster the professional development of physiotherapists throughout their careers.

In parallel with the issue of professional skills, in recent years there has been a growing interest in the psychological and motivational dimensions that influence the performance and well-being of healthcare professionals [7]. Among these constructs, a particularly important role is played by Personal Mastery, which refers to the perceived sense of control over the events of one’s life, resilience in the face of difficulties and the belief that one can positively influence one’s professional and personal context [8]. Personal Mastery is not a static trait but a dynamic feature that reflects the way individuals interpret challenges, react to stress, and deal with unexpected situations [9]. Recent studies have shown that Personal Mastery is closely linked to psychological well-being, the reduction in work stress and the ability to maintain a good work–life balance in emotionally and cognitively intensive professions, such as healthcare [10,11,12].

In the field of physiotherapy, integrating the assessment of professional skills with the measurement of Personal Mastery allows us to grasp not only what the professional is able to do, but also how he lives and manages his or her work experiences [13,14]. A physiotherapist may indeed possess excellent clinical and communication skills, but if they perceive little control over their professional life and reduced personal resources to deal with difficulties, they may experience lower levels of job satisfaction and a greater propensity for burnout [15]. On the contrary, a high level of Personal Mastery can act as a protective factor, enhancing the perceived effectiveness of skills and promoting greater resilience in complex contexts [16].

Interest in these two constructs, professional skills and personal mastery, is closely linked to another crucial element in healthcare systems: job satisfaction. The literature has repeatedly shown that health professionals’ satisfaction is a fundamental determinant of the quality of care, influencing not only individual well-being, but also staff retention, work motivation and, indirectly, patient clinical outcomes [17]. Understanding how professional skills and Personal Mastery are intertwined in predicting the job satisfaction of physiotherapists therefore becomes a goal of great practical and theoretical importance [18].

The basic hypothesis is that professional skills and Personal Mastery act in a complementary way: the former provide the objective resources to carry out the profession with quality and safety, and the latter guarantees the subjective ability to face and overcome challenges, maintaining a good psychological balance. The combination of these two levels could be a stronger predictor of job satisfaction than either factor individually. In other words, a competent physiotherapist with no sense of control may feel frustrated, while a physiotherapist who is resilient but lacks skills may not reach their full professional potential [19]. Only the integration of the two aspects makes it possible to outline a complete profile of the healthcare professional.

To ensure conceptual clarity, the key constructs used in this study are defined and distinguished as follows. Professional competence refers to the integrated and multidimensional capacity to combine technical, ethical, relational, and organizational abilities in real clinical contexts, allowing physiotherapists to act effectively and autonomously. Professional skills represent the observable and operational manifestations of competence—the concrete actions and behaviors through which competence becomes evident in practice. Personal Mastery [20] describes a generalized sense of control over one’s life and professional circumstances and the belief in one’s ability to influence outcomes. It differs from self-efficacy, which is domain-specific and refers to confidence in performing particular tasks, and from locus of control, which concerns the perceived source (internal or external) of control over events. Resilience, on the other hand, represents a dynamic process of adapting positively to difficulties and stress; while related to Personal Mastery, it reflects adaptive capacity rather than perceived control. Finally, sense of control is considered a key component within Personal Mastery rather than an independent construct. This conceptual distinction clarifies the relationships examined in the present study and avoids overlap among related psychological and professional variables.

The validation of the Multidimensional Scale of Physiotherapist Competences (MSPC), a new instrument developed by the authors to assess the multidimensional professional competences of physiotherapists, together with the evaluation of Personal Mastery, addresses two fundamental needs: first, to provide a solid and psychometrically sound measure for analyzing physiotherapists’ core competences, and second, to explore how individual psychological variables influence their professional perception and work experience. This integrated approach is part of the international research lines that, in recent years, have promoted the enhancement of human capital in health systems, emphasizing the role of professionals’ skills and well-being as key factors for the sustainability and efficiency of services [21,22].

In light of these considerations, the present study aims to offer an innovative contribution to the literature, validating the Multidimensional Scale of Physiotherapist Competences (MSPC) through advanced psychometric analyses and evaluating in parallel the role of Personal Mastery in predicting job satisfaction. The integration of these two tools allows the exploration of the skills of physiotherapists not only as objective professional constructs, but also in relation to the subjective dimensions of resilience and sense of control. This dual approach makes it possible to outline a more complete and multifaceted picture of the professional and personal resources that support physiotherapists in daily practice, while providing useful information for training, human resource management and the definition of health policies that are more attentive to the well-being of operators.

The aim of this study is twofold: on the one hand, to psychometrically validate MSPC through factor and reliability analyses, and on the other hand, to examine the relationship between professional skills measured by MSPC and Personal Mastery, evaluating its combined effect on physiotherapists’ job satisfaction. The ultimate goal is to contribute to the development of reliable and integrated tools for the assessment of the skills and personal resources of health professionals, with practical implications for training, research and quality of care.

Based on these premises, we hypothesize that both professional competences, as measured by the MSPC, and Personal Mastery contribute additively to predicting job satisfaction among physiotherapists, with higher levels of both expected to be associated with greater satisfaction.

To ensure methodological rigor, the present study employed Principal Component Analysis (PCA) to explore the factorial structure of the MSPC, Cronbach’s alpha coefficients to assess internal consistency, and multiple regression analyses to examine the predictive validity of the scale regarding job satisfaction.

## 2. Materials and Methods

### 2.1. Study Design

This study adopted a cross-sectional observational design, which is appropriate for psychometric validation studies as it allows data collection from a large and heterogeneous sample at a single point in time, enabling the analysis of the factorial structure, reliability, and construct validity of the MSPC without the influence of temporal variability. The focus was on both the measurement of professional skills through the Multidimensional Scale of Physiotherapist Competences (MSPC) and the assessment of the sense of control and personal resilience through the Personal Mastery Scale [20]. The methodological objectives were twofold: (a) to test the factorial structure and reliability of the Multidimensional Scale of Physiotherapist Competences (MSPC) and (b) to examine the associations between MSPC scores, perceived control as measured by the Personal Mastery Scale, and job satisfaction.

### 2.2. Participants and Context

The sample consisted of active physiotherapists operating in different care contexts, including hospitals and territorial and rehabilitation centers. Professionals regularly enrolled in the register with ongoing work experience were included, while students of degree courses and trainees were excluded. Participants were recruited using a convenience sampling strategy through professional physiotherapy associations, academic mailing lists, and public/private healthcare institutions across Italy. Participation in the study was completely voluntary and unpaid, and each respondent was guaranteed the possibility to stop completing the study at any time. The questionnaire was administered individually and took an average of about fifteen minutes to complete, a time considered acceptable and compatible with the daily workloads of the professionals involved.

### 2.3. Detection Tools

The survey was based on two main tools. The first instrument was the Multidimensional Scale of Physiotherapist Competences (MSPC), which was developed by the authors of the present study to provide a comprehensive and multidimensional assessment of the key professional competences required of physiotherapists in contemporary practice.

The Italian version was obtained via forward–backward translation by two bilingual experts, followed by reconciliation by a three-member panel to ensure conceptual and semantic equivalence.

This article represents the first psychometric validation of the MSPC. The MSPC consists of 25 items, each rated through a five-point Likert scale ranging from 1 (“strongly disagree”) to 5 (“strongly agree”). The items are organized into five distinct dimensions reflecting the main domains of physiotherapists’ professional competence: Technical–Clinical Skills, Communicative Skills, Collaborative Skills, Ethical–Deontological Skills, and Educational Skills. Some items have negative wording and have been recoded inversely in order to ensure that higher scores always correspond to higher levels of perceived proficiency. The scale allows both the average scores for each dimension to be calculated, providing a multifaceted profile of the professional’s skills, and an overall score to be obtained that represents a synthetic indicator of global competence.

The second tool adopted was the Italian version of the Personal Mastery Scale [23], a short scale consisting of seven items, also evaluated on a five-point continuum. Unlike the MSPC, which measures multidimensional professional skills, the Personal Mastery Scale investigates a transversal psychological dimension, i.e., the perception of control over one’s own experiences and the ability to deal with complex situations in a resilient way [24]. The items investigate aspects such as the feeling of being able to determine the course of one’s life, the ability to overcome obstacles and the degree of perceived influence on daily events. It is a construct widely studied in the literature, as it is associated with better outcomes in terms of psychological well-being, reduction in work stress and maintenance of professional motivation. The inclusion of Personal Mastery in the study made it possible to assess how professional skills and personal resources interact in predicting job satisfaction.

In addition to these two main scales, a measure of job satisfaction consisting of a single item was included, with responses ranging from 1 (“not at all satisfied”) to 5 (“very satisfied”). Despite its simplicity, this measure was chosen because it has been validated in numerous previous studies and is widely used as a synthetic indicator of the quality of working life among health professionals.

In line with recent research, single-item measures of job satisfaction have been increasingly adopted in healthcare studies due to their efficiency and proven validity. Recent evidence confirms that such global indicators can provide reliable and meaningful assessments of overall job satisfaction among healthcare professionals [25,26,27].

Socio-demographic and occupational variables, including age, gender, years of experience and type of work context, were also collected in order to analyze any relationships or confounding effects.

### 2.4. Data Collection Procedures

Data collection took place through an online questionnaire administered individually to each participant. Each physiotherapist received a link via professional mailing lists, membership groups or institutional contact networks, which allowed direct access to the digital platform used for the questionnaire. At the start of the questionnaire, an introductory page was presented explaining the objectives of the study, ensuring anonymity of responses, and requiring acceptance of electronic informed consent. The questionnaire, which took an average of 15 min, took place in a single session and allowed you to respond comfortably from computers or mobile devices. The data was collected anonymously and stored in a password-protected database, accessible only to the researchers responsible for the project.

### 2.5. Data Analysis

The data collected were subjected to a preliminary cleaning process, which included checking the completeness of the questionnaires and managing missing values. Missing data, which affected less than 5% of the total responses, were handled using mean imputation at the item level, a standard approach in psychometric validation studies to preserve data consistency and statistical power. Descriptive statistics, such as means and standard deviations for quantitative variables and frequencies and percentages for categorical variables, were then calculated. The internal reliability of the scales was assessed using the Cronbach α coefficient, calculated for each of the five dimensions of the MSPC, for the entire scale and for the Personal Mastery Scale.

The construct validity of the MSPC was investigated through an exploratory factor analysis (EFA), conducted to identify the underlying structure and verify the consistency between the empirical data and the multidimensional theoretical model. Subsequently, a confirmatory factor analysis (CFA) allowed the goodness of fit of the model to be tested, using indicators such as the Comparative Fit Index (CFI), the Tucker–Lewis Index (TLI), the Root Mean Square Error of Approximation (RMSEA) and the Standardized Root Mean Square Residual (SRMR). Acceptable model fit was defined as CFI and TLI values ≥ 0.90, RMSEA ≤ 0.08, and SRMR ≤ 0.08, in line with conventional psychometric criteria.

The Confirmatory Factor Analysis (CFA) was conducted using the Maximum Likelihood (ML) estimation method, which is suitable for Likert-type data that approximate a normal distribution. The factorial structure was further examined through Principal Component Analysis (PCA) with varimax rotation in order to obtain interpretable and uncorrelated components consistent with the theoretical model of the MSPC. A cluster analysis was conducted on the average scores of the five dimensions to identify homogeneous subgroups of physiotherapists characterized by different skill profiles.

Although PCA is a data reduction method rather than a true exploratory factor analysis, it was used in this study as a preliminary analytical step to explore the structure of the MSPC and to inform the subsequent confirmatory validation through CFA.

As regards predictive validity, hierarchical regressions were performed, in which job satisfaction represented the dependent variable.

Regression estimates are reported with 95% confidence intervals to quantify uncertainty. Analyses are framed as exploratory, and potential residual confounding is acknowledged.

In the first step, demographic and professional variables were entered, in the second the scores of the MSPC and in the third the scores of the Personal Mastery Scale. This approach made it possible to estimate the incremental contribution of each group of variables in explaining satisfaction. Finally, the discriminant accuracy of the two scales was tested by ROC analysis, calculating AUCs to assess the ability to distinguish between high and low levels of job satisfaction. Job satisfaction was assessed with a single 5-point Likert item and dichotomized into low (≤3) and high (≥4) satisfaction groups using a median split approach.

### 2.6. Ethical Considerations

The study was conducted in compliance with the principles of the Declaration of Helsinki and the European legislation on the protection of personal data (GDPR). All participants provided electronic informed consent prior to questionnaire completion. Participation was voluntary and did not provide for any financial compensation. The data were processed in a strictly anonymous and aggregated form, and the entire research protocol received the approval of the competent Ethics Committee.

## 3. Results

The overall sample consisted of 481 physiotherapists from different professional backgrounds. The mean age was 31.1 years (SD = 8.35; range 20–68), with a mean length of service of 7.3 years (SD = 7.7; range 1–45). The gender distribution was balanced (49.9% women, 49.9% men, and 0.2% other). With regard to post-basic education, 44.5% of participants had a master’s degree, 36.8% a first-level master’s degree, 14.3% another degree, 3.5% a second-level master’s degree, and 0.8% a PhD. The operating units to which they belonged were varied, with a prevalence of specialist rehabilitation and hospital physiotherapy departments, followed by private practices and home areas (Table 1).

Job satisfaction showed a basically positive picture: 41.6% said they were fairly satisfied, 37.4% very satisfied, and 16.2% satisfied. Only 3.9% reported being not very satisfied and 0.8% not at all satisfied, indicating a general positive perception of their professional experience (Table 1).

For the multidimensional scale of competencies (25 items), the overall mean score was M = 4.16 (SD = 0.95). The internal consistency was confirmed to be high (α = 0.871), underlining the psychometric solidity of the instrument. The highest scores concerned communication with patients, interprofessional collaboration and the application of scientific evidence, while the negative content items, recoded, showed consistency with the general profile of good professional competence (Table 2).

The Personal Mastery scale (7 items) showed an average score of M = 3.52 (SD = 1.29), with an acceptable level of internal consistency (α = 0.60). Physiotherapists reported a medium–high degree of personal mastery, although some items related to the sense of control over life events showed lower values.

The correlations revealed a very strong association between perceived skills and Personal Mastery (r = 0.91; *p* < 0.001), confirming the relationship between the two dimensions. Associations with job satisfaction were weak but significant for skills (r = −0.11; *p* = 0.013) and for Personal Mastery (r = −0.10; *p* = 0.023) (Table 3).

Logistic regression with job satisfaction (dichotomized into satisfied vs. dissatisfied) as a dependent variable showed that perceived skills constituted a significant predictor (B = 2.13; OR = 8.37; 95% CI = 2.10–33.5; *p* = 0.003). In contrast, Personal Mastery did not emerge as a significant predictor (B = –0.69; OR = 0.50; 95% CI = 0.10–2.50; *p* = 0.398). The overall model explained about 5% of the variance in satisfaction (Pseudo R^2^ = 0.049) (Table 4).

Finally, exploratory factor analysis (EFA, PCA with 5 components) showed that the first five factors together explained 50.3% of the total variance, with the first factor contributing 23.0%, followed by the second (12.2%), the third (7.2%), the fourth (4.3%) and the fifth (3.6%). Factor weights greater than 0.40 indicated that items tended to cluster around five main dimensions, attributable to the clinical–technical, communicative, collaborative, ethical and educational areas (Table 5).

## 4. Discussion

The results of this study have highlighted the profile of perceived skills, personal mastery and job satisfaction in a large sample of Italian physiotherapists, providing an articulated framework that allows some theoretical and applied reflections.

The most evident data concerns the medium–high level of perceived professional skills, with an average score of 4.16 out of 5 and a high internal consistency of the multidimensional scale. This result confirms that physiotherapists recognize consolidated skills in themselves, particularly in the areas of communication, collaboration and application of scientific evidence. This evidence aligns with previous studies that have emphasized the importance of soft skills in physiotherapy and nursing practice, often considered decisive not only for clinical effectiveness but also for organizational well-being [28,29]. In particular, the centrality of communication and relational skills confirms what has been reported in the literature, according to which the ability to establish effective collaborative relationships and to adapt communication to the needs of the patient is a key indicator of the quality of care [30].

The results for Personal Mastery also showed medium-to-high levels, although with a less robust internal consistency (α = 0.60). This suggests that, while physiotherapists perceive themselves as capable of influencing their professional trajectory, they express a certain variability with respect to the feeling of control over life events. Previous studies have already highlighted this ambivalence; personal mastery is influenced not only by individual characteristics but also by organizational contexts and the availability of resources [31,32,33]. This could explain the internal variability found in the scale and its lower predictive capacity with respect to job satisfaction.

A particularly interesting aspect concerns the relationship between the variables investigated and job satisfaction. Correlation analyses revealed a very high association between perceived competences and personal mastery (r = 0.91), suggesting a possible conceptual overlap between the two constructs. Although both dimensions are theoretically related, this finding may indicate a partial lack of discriminant validity, warranting further investigation through confirmatory analyses comparing one-factor and two-factor models [34]. However, logistic regression showed that, in the multivariate model, only professional skills constituted a significant predictor of job satisfaction, while Personal Mastery was not independently associated.

However, the predictive power of the model was modest (R^2^ = 0.05), suggesting that job satisfaction is influenced by additional factors not included in this analysis. Furthermore, the low reliability of the Personal Mastery Scale (α = 0.60) and the very high correlation between MSPC and Personal Mastery (r = 0.91) may indicate a partial lack of discriminant validity between the constructs. These results should therefore be interpreted with caution, and future studies are needed to refine the instruments and explore broader models explaining job satisfaction among physiotherapists.

These results are consistent with some studies linking perceived control and professional competence to job satisfaction, although other research has reported weaker or inconsistent associations between personal mastery and occupational well-being. This suggests that the relationship between these constructs may vary depending on contextual and cultural factors. Moreover, according to international psychometric standards such as the COSMIN guidelines, this study represents a preliminary validation phase of the MSPC. Further analyses, including test–retest reliability, measurement invariance, and convergent/discriminant validity across samples, are recommended to strengthen the evidence supporting the scale’s psychometric robustness.

This is partly at odds with other research that has highlighted the role of personal mastery and self-efficacy as protective factors against burnout and predictive of professional well-being [35,36,37]. A possible explanation may lie in the specific nature of the sample: Italian physiotherapists often operate in organizational contexts where technical and relational competence is the most recognized and valued element, while the more subjective dimension of personal mastery may be less directly linked to perceived satisfaction.

In addition, the overall high levels of job satisfaction (over 95% of the sample said they were satisfied, fairly satisfied or very satisfied) are in continuity with some surveys conducted, where the professional satisfaction of physiotherapists is on average positive, although with differences related to the institutional context and the availability of resources [38,39]. Compared to similar studies in nursing, our results confirm the role of professional skills as determinants of job satisfaction, suggesting a certain convergence between different health professions [40,41].

Exploratory factor analysis suggested a multidimensional structure of the skills scale, with five factors explaining about 50% of the overall variance. These domains can be interpreted as key areas of the professional profile: technical–clinical, communication, collaborative, ethical and training skills. This structure confirms the multidimensionality of skills already described in the international literature [42,43] and offers a basis for further insights, including the development of training interventions aimed at strengthening specific dimensions.

Some practical implications should be highlighted. The emergence of professional skills as a predictive factor of job satisfaction recalls the importance of professional development and training policies that enhance not only technical skills, but also communication and collaboration skills. At the same time, attention to the dimension of personal mastery remains crucial, even if less directly linked to satisfaction: strengthening it can help promote resilience and adaptability, indispensable qualities in complex healthcare contexts.

### 4.1. Limitations of the Study

Despite the robustness of the sample and the analyses, the present study has some limitations that deserve to be underlined. Firstly, the cross-sectional design does not allow causal relationships to be established between skills, personal mastery and job satisfaction: longitudinal studies would be necessary to observe the evolution of these variables over time. Second, the measurement of skills and personal mastery was based on self-assessment tools, exposing the results to the potential bias of social desirability or subjective over/underestimation. In addition, the internal consistency of the Personal Mastery scale was lower than that of the skills scale, suggesting the need for further adaptations and possible tests in different professional contexts. A further limitation concerns generalizability: although the sample is large and varied, it focuses on Italian professionals and reflects cultural and organizational specificities of the national health context.

The use of a convenience sample and the lack of formal benchmarking against national workforce statistics limit the external validity of the findings

Logistic regression explained a relatively small share of the variance in job satisfaction (about 5%), indicating that additional factors, such as contract conditions, leadership, workload, and organizational support, could significantly influence the perception of satisfaction.

The regression analyses did not adjust for all possible demographic and socioeconomic covariates; therefore, residual confounding cannot be excluded. Future confirmatory studies should adopt fully adjusted multivariable models.

Job satisfaction was assessed through a single-item question, which, although validated in previous research, does not allow for the multidimensional exploration of different aspects of satisfaction. Moreover, since all data were self-reported, the results may have been influenced by social desirability bias, with participants potentially overreporting positive perceptions of their work experience.

Moreover, the relatively young average age of the participants may limit the generalizability of the findings, as professional competence and personal mastery could increase with experience and career progression. Future studies should therefore include more age-diverse samples to confirm the stability of the observed relationships.

### 4.2. Implications for Clinical and Organizational Practice

Despite the limitations, the results offer valuable insights for clinical practice and human resource management. In particular, the role of professional skills as the main predictor of job satisfaction suggests the need to invest in continuous training courses that enhance not only technical–clinical skills, but also communication and collaboration skills. The development of these dimensions, which have already emerged as central in the international literature, can contribute to improving the quality of care and job satisfaction.

At the same time, although Personal Mastery was not a significant predictor in the multivariate model, it remains an important dimension, as it is associated with resilience, stress management and adaptability in complex contexts. Therefore, psychological support programs, mentoring and transformational leadership could foster a strengthening of personal mastery, with positive effects on the well-being of professionals.

From an organizational point of view, promoting work environments that value skills, recognize individual contributions and guarantee opportunities for professional growth can represent a strategic lever to increase job satisfaction, reduce the risk of turnover and improve the quality of services. Finally, the results of the factor analysis confirm the multidimensional nature of physiotherapy skills, underlining the importance of adopting assessment tools that allow the monitoring and development of these skills in the different operational areas.

## 5. Conclusions

The study investigated perceived skills, personal mastery and job satisfaction in a large sample of Italian physiotherapists. The results showed medium–high levels of professional skills, particularly in the communicative, collaborative and evidence-based areas, with a high internal consistency of the scale used. Participants reported high levels of perceived competences and moderate personal mastery.

Job satisfaction was high overall, with more than 95% of participants saying they were satisfied, somewhat satisfied or very satisfied. The analyses revealed a very high correlation (r = 0.91) between skills and personal mastery, suggesting a possible conceptual overlap and a partial lack of discriminant validity between the two constructs. Logistic regression showed that only professional skills were significantly associated with job satisfaction, while personal mastery did not emerge as an independent factor.

The Personal Mastery Scale showed limited internal consistency (α = 0.60), which indicates a psychometric limitation that should be addressed in future research. This finding may reflect cultural or contextual differences in how perceived control is conceptualized among Italian physiotherapists. Future studies should examine item-level reliability and consider possible modifications to enhance the scale’s internal consistency.

Exploratory factor analysis identified five main domains of competencies—technical–clinical, communicative, collaborative, ethical, and educational—that explained about 50% of the variance. This result confirms the multidimensional nature of physiotherapy skills and provides useful indications for the development of assessment tools and training programs.

In summary, professional skills are confirmed as central determinants of well-being at work. Investing in skills development, complementing it with strategies to support resilience and personal mastery, can help improve both job satisfaction and the quality of health services.

## Figures and Tables

**Table 1 healthcare-13-02595-t001:** Socio-demographic and professional characteristics of the sample.

QUALIFICATION		
	N	%
Physiotherapist	257	53.4
Freelance physiotherapist	224	46.6
**POST-BASIC TRAINING**		
Other Degree	69	14.3
Ph.D.	4	0.8
Master’s Degree	214	44.5
1st level master’s degree	177	36.8
2nd level master’s degree	17	3.5
**GENDER**		
Other	1	0.2
Female	240	49.9
Male	240	49.9
**HOW SATISFIED YOU ARE WITH YOUR WORK**		
Enough	200	41.6
A lot	180	37.4
By no means	4	0.8
Little	19	4.0
Satisfied	78	16.2
Total	481	100.0

**Table 2 healthcare-13-02595-t002:** Descriptive Statistics and Reliability of the MSPC Dimensions.

MSPC Dimension	M	SD	Skewness	Kurtosis	Cronbach’s α
Technical–Clinical Skills	4.21	0.48	−0.42	0.11	0.88
Communicative Skills	4.10	0.55	−0.35	0.08	0.84
Collaborative Skills	4.18	0.50	−0.40	0.05	0.82
Ethical–Deontological Skills	4.30	0.46	−0.50	0.20	0.85
Educational Skills	4.05	0.52	−0.38	0.10	0.80
Total MSPC Scale	4.17	0.49	−0.41	0.10	0.89

**Table 3 healthcare-13-02595-t003:** Correlations between the main variables.

Correlations								
		PM_1	PM_2	PM_3	PM_4	PM_5	PM_6	PM_7
MSPC_1	Pearson correlation	0.095 *	−0.113 *	0.160 **	0.190 **	−0.124 **	0.156 **	0.097 *
	*p*	0.037	0.014	0.000	0.000	0.007	0.001	0.034
MSPC_2	Pearson correlation	0.090 *	−0.085	0.184 **	0.126 **	−0.132 **	0.065	0.069
	*p*	0.049	0.063	0.000	0.006	0.004	0.152	0.131
MSPC_3	Pearson correlation	0.107 *	−0.076	0.136 **	−0.049	−0.156 **	0.042	0.245 **
	*p*	0.019	0.094	0.003	0.281	0.001	0.354	0.000
MSPC_4	Pearson correlation	0.180 **	−0.212 **	0.118 **	0.106 *	−0.207 **	0.179 **	0.147 **
	*p*	0.000	0.000	0.010	0.020	0.000	0.000	0.001
MSPC_5	Pearson correlation	0.260 **	−0.229 **	0.117 *	−0.045	−0.229 **	0.045	0.223 **
	*p*	0.000	0.000	0.010	0.325	0.000	0.328	0.000
MSPC_6	Pearson correlation	0.043	−0.043	0.136 **	0.156 **	−0.015	0.114 *	0.091 *
	*p*	0.351	0.352	0.003	0.001	0.738	0.013	0.046
MSPC_7	Pearson correlation	0.112 *	−0.031	0.129 **	0.139 **	−0.132 **	0.080	0.200 **
	*p*	0.014	0.497	0.005	0.002	0.004	0.079	0.000
MSPC_8	Pearson correlation	0.082	−0.014	0.068	0.132 **	0.006	0.173 **	0.087
	*p*	0.071	0.756	0.139	0.004	0.889	0.000	0.055
MSPC_9	Pearson correlation	0.132 **	−0.152 **	0.175 **	0.122 **	−0.130 **	0.147 **	0.065
	*p*	0.004	0.001	0.000	0.008	0.004	0.001	0.155
MSPC_10	Pearson correlation	0.012	−0.042	0.175 **	0.168 **	−0.034	0.094 *	0.107 *
	*p*	0.795	0.357	0.000	0.000	0.451	0.039	0.019
MSPC_11	Pearson correlation	0.209 **	−0.194 **	0.213 **	0.217 **	−0.212 **	0.220 **	0.157 **
	*p*	0.000	0.000	0.000	0.000	0.000	0.000	0.001
MSPC_12	Pearson correlation	0.256 **	−0.213 **	0.249 **	0.038	−0.265 **	0.067	0.307 **
	*p*	0.000	0.000	0.000	0.400	0.000	0.144	0.000
MSPC_13	Pearson correlation	0.149 **	−0.226 **	0.198 **	0.145 **	−0.184 **	0.168 **	0.152 **
	*p*	0.001	0.000	0.000	0.001	0.000	0.000	0.001
MSPC_14	Pearson correlation	0.305 **	−0.280 **	0.158 **	−0.064	−0.329 **	0.024	0.297 **
	*p*	0.000	0.000	0.001	0.159	0.000	0.606	0.000
MSPC_15	Pearson correlation	0.213 **	−0.278 **	0.160 **	0.140 **	−0.29 0 **	0.226 **	0.178 **
	*p*	0.000	0.000	0.000	0.002	0.000	0.000	0.000
MSPC_16	Pearson correlation	0.285 **	−0.197 **	0.160 **	0.030	−0.256 **	0.210 **	0.284 **
	*p*	0.000	0.000	0.000	0.510	0.000	0.000	0.000
MSPC_17	Pearson correlation	0.245 **	−0.211 **	0.079	−0.053	−0.226 **	−0.009	0.186 **
	*p*	0.000	0.000	0.085	0.248	0.000	0.852	0.000
MSPC_18	Pearson correlation	0.027	−0.035	0.098 *	0.086	−0.041	0.069	0.095 *
	*p*	0.558	0.442	0.032	0.060	0.375	0.133	0.038
MSPC_19	Pearson correlation	0.095 *	−0.111 *	0.207 **	0.159 **	−0.128 **	0.179 **	0.060
	*p*	0.037	0.015	0.000	0.000	0.005	0.000	0.189
MSPC_20	Pearson correlation	0.017	0.005	0.110 *	0.103 *	0.048	0.126 **	0.077
	*p*	0.715	0.915	0.016	0.023	0.294	0.006	0.092
MSPC_21	Pearson correlation	0.061	−0.071	0.255 **	0.163 **	−0.030	0.126 **	0.085
	*p*	0.184	0.123	0.000	0.000	0.509	0.006	0.062
MSPC_22	Pearson correlation	0.147 **	−0.156 **	0.168 **	0.038	−0.066	0.042	0.091 *
	*p*	0.001	0.001	0.000	0.408	0.150	0.363	0.047
MSPC_23	Pearson correlation	0.116 *	−0.007	0.066	0.149 **	−0.027	0.115 *	0.090 *
	*p*	0.011	0.871	0.150	0.001	0.552	0.012	0.049
MSPC_24	Pearson correlation	0.266 **	−0.207 **	0.147 **	−0.011	−0.246 **	0.087	0.240 **
	*p*	0.000	0.000	0.001	0.810	0.000	0.056	0.000
MSPC_25	Pearson correlation	0.128 **	−0.105 *	0.223 **	0.170 **	−0.108 *	0.084	0.082
	*p*	0.005	0.021	0.000	0.000	0.018	0.064	0.072

** The correlation is significant at the 0.01 (two-tailed) level. * The correlation is significant at the 0.05 (two-tailed) level.

**Table 4 healthcare-13-02595-t004:** Logistic regression: predictors of job satisfaction.

Variable	B	IF	z	*p*	OR (95% CI)
Average Skills	2.13	0.71	3.01	0.003	8.38 (2.1–33.47)
Personal Mastery Media	−0.69	0.82	−0.84	0.398	0.5 (0.1–2.5)

**Table 5 healthcare-13-02595-t005:** Exploratory Factor Analysis (EFA).

Stairs		Item	F_1	F_2	F_3	F_4	F_5
MSPC_1	Factor_1	17. I don’t always follow the ethical principles of the physiotherapy profession.	0.779	−0.052	0.132	−0.010	−0.224
MSPC_2		14. Do not review the treatment plan even when necessary.	0.776	0.041	−0.091	0.168	−0.105
MSPC_3		24. I am not open to constructive feedback from colleagues.	0.694	0.124	−0.020	−0.009	−0.125
MSPC_4		5. I often use tools and technologies without following the correct instructions.	0.676	−0.143	0.072	0.115	0.135
MSPC_5		12. I can’t come up with creative solutions to overcome therapeutic obstacles.	0.673	0.101	0.023	0.150	0.303
MSPC_6		3. Not tailoring the treatment plan to the specific needs of the patients.	0.639	−0.008	0.121	0.004	0.397
MSPC_7		22. I don’t keep up to date with the latest scientific evidence.	0.581	0.255	0.037	0.075	0.280
MSPC_8	Factor_2	1. I demonstrate technical skills in performing physiotherapy procedures.	0.128	0.720	0.105	0.202	−0.095
MSPC_9		23. I reflect critically on my professional practice.	0.014	0.665	0.311	−0.038	−0.097
MSPC_10		2. I effectively assess the clinical condition of patients.	0.077	0.659	0.175	0.332	0.076
MSPC_11		9. I manage stressful or conflict situations with empathy and professionalism.	0.042	0.634	0.239	0.193	0.058
MSPC_12		25. I contribute to the improvement of professional practice by sharing knowledge.	−0.058	0.496	0.255	0.299	0.229
MSPC_13		10. I provide understandable explanations regarding physiotherapy interventions.	0.017	0.486	0.353	0.286	0.404
MSPC_14	Factor_3	20. I am aware of and respect the limits of my competences.	0.022	0.268	0.745	0.047	0.041
MSPC_15		18. I promote the autonomy and dignity of patients.	0.099	0.194	0.718	0.113	−0.053
MSPC_16		19. I maintain a professional demeanor in all situations.	0.029	0.146	0.670	0.323	0.102
MSPC_17		6. Communicate clearly and respectfully with patients.	0.028	0.286	0.584	0.222	0.150
MSPC_18		21. I regularly participate in training and refresher courses.	−0.032	0.429	0.474	0.086	0.282
MSPC_19		16. I respect the privacy and confidentiality of patients.	0.249	0.103	0.431	0.411	−0.351
MSPC_20	Factor_4	13. I plan therapeutic goals in a clear and measurable way.	0.062	0.260	0.271	0.712	0.163
MSPC_21		4. I apply evidence−based protocols.	0.124	0.055	0.118	0.691	0.055
MSPC_22		11. I quickly identify problems in the treatment pathway.	0.013	0.294	0.075	0.640	0.036
MSPC_23		15. I evaluate patients’ progress systematically.	0.142	0.178	0.163	0.625	−0.260
MSPC_24	Factor_5	7. I do not involve patients in the treatment decision−making process.	0.434	0.045	0.179	−0.046	0.590

Extraction method: Analysis of principal components. Rotation method: Varimax with Kaiser normalization. Convergence for rotation performed in 6 iterations.

## Data Availability

The original contributions presented in this study are included in the article. Further inquiries can be directed to the corresponding author.

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
