# Peer review of "Professional Competencies and Job Satisfaction Among Physiotherapists: Validation and Psychometric Analysis of the Multidimensional Scale"

_healthcare, 2025, doi:10.3390/healthcare13202595_

Round 1

Reviewer 1 Report

Comments and Suggestions for Authors

Corrections should be made:
Abstract:
- line 28 - expand shortcut PCA
Introduction:
- has the MSPC been validated previously? In what language/country? Is there no scale that integrates competence and personal mastery for physiotherapists? Be explicit.
-  you refer to “tools such as the MSPC” (line 103) but never say what MSPC is, who developed it, how many items/dimensions, or what previous psychometric evidence exists. The reader must know the scale’s origin and why it needs re-validation here - please provide brief and concise description
-  you repeatedly state that competence and Personal Mastery combine to predict job satisfaction, but you never define whether you expect additive, moderation or mediation. The paper must state testable hypotheses (add it at the and of introduction section)
-professional skills,” “competence,” “Personal Mastery,” “resilience,” “sense of control” are used interchangeably which disturbs the reader. Define each term precisely and explain how they differ from related constructs (maybe self-efficacy, locus of control, resilience)
- For a psychometric validation papers usually the Introduction section should briefly state what psychometric methods will be used (EFA/CFA, reliability indices, measurement invariance, SEM/regression) and why 
Methods:
- line 130 - “Transversal observational design”. The standard term is “cross-sectional observational design.” - please correct
-  text justifies cross-sectional design with cost/speed reasons; for psychometric validation you must emphasize why cross-sectional is appropriate for factor analysis and construct validity (not “representative picture” or “reduced costs”).  
-  you write “assessment of the sense of control and personal resilience through the Personal Mastery Scale” which is inaccurate. The Personal Mastery Scale measures perceived control, not resilience. Do not conflate constructs.  - the sentence “Methodological objective was twofold” is vague. You need to state explicitly: like that: (a) test the factor structure and reliability of MSPC; (b) examine the association between MSPC scores, Personal Mastery, and job satisfaction.
- you should avoid “contribution of personal mastery in explaining professional satisfaction” because it implies causal inference, which is not supported by cross-sectional data. Use “association with job satisfaction” instead.
2.3 section:
-  you don’t provide reference(s) for the MSPC. A validation paper must cite the original development source.
- “Scale developed with the intention of comprehensively capturing…” is too vague. You need a precise description: who developed it, in what context, original validation results - describe shortly
- “25 items” and “five dimensions” are fine, but the exact names of the subscales should match the original authors’ terminology (yours differ slightly, e.g., “ethical and deontological skills”).

data analysis section:
- “AFE” and “AFC” are not standard in English (they’re Spanish/Italian acronyms). Must use EFA (Exploratory Factor Analysis) and CFA (Confirmatory Factor Analysis).
- “Constructive validity” . Correct term is construct validity.
- Current description (“averaging cases with low missing rates”) is vague and statistically questionable. You must specify how missing data were treated (e.g., listwise deletion, mean imputation, expectation-maximization, multiple imputation). Averaging across cases is not acceptable unless justified.
- Need to define what “low missing rates” means (e.g., <5%???)
- No mention of extraction method (e.g., principal axis factoring, maximum likelihood) or rotation (e.g., oblimin/varimax). Must specify.
- For CFA: no mention of estimator (e.g., ML, WLSMV if data are ordinal Likert). 
- Need to state cut-off criteria for CFI, TLI, RMSEA, SRMR..- Using ROC/AUC for “high vs low job satisfaction” is problematic because job satisfaction was measured with a single Likert item (ordinal, not binary). You need to specify how you dichotomized (maybe median split, cut-off at ≥4).

Results:
- the correlation of r = 0.91 between MSPC and Personal Mastery is extremely high, suggesting multicollinearity or that both scales are essentially measuring the same construct (lack of discriminant validity). This undermines the rationale for including both tools and requires additional testing (e.g., CFA comparing one-factor vs. two-factor models).
- Low reliability of the Personal Mastery Scale (Cronbach’s α = 0.60 is below the accepted psychometric threshold (≥0.70). In its current form, the scale does not meet reliability standards. Item-level diagnostics should be reported (e.g., item–total correlations, α if item deleted), and modifications or exclusion of weak items should be considered)
- Table 2 incorrectly reports Cronbach’s α per item (MSPC_1–25)  α should only be reported for the whole scale or its subscales!!!!!!!!!!!! Descriptive statistics are incomplete (no skewness, kurtosis, or item-level distributions).
- The study used PCA instead of true exploratory factor analysis (EFA) — an inappropriate method for psychometric validation.

Discussion: 
- The discussion treats the MSPC as highly reliable, while in fact the Personal Mastery scale had very low reliability (α = 0.60) and this is downplayed. Next: the extremely high correlation (r = 0.91) between MSPC and Personal Mastery is interpreted as “mutually reinforcing,” instead of being flagged as a serious validity problem (lack of discriminant validity). Moreover, logistic regression results are overemphasized, despite the fact that the model explained only 5% of variance in job satisfaction.
- litterature: Citations are used mostly to confirm the findings, but not critically — e.g., conflicting evidence on personal mastery and job satisfaction is mentioned but not explored in depth. No reference to psychometric literature (e.g., COSMIN, validation standards) to address problems with scale reliability, validity, or factor analysis.

Conclusions:
- you write the Personal Mastery scale “acceptable internal consistency,” while in fact α = 0.60 is weak, not acceptable by most psychometric standards.
- The very high correlation between MSPC and Personal Mastery is described as a “strong relation,” but not problematized as a validity issue (construct overlap, lack of discriminant validity).

Author Response

Dear Reviewer 1,

Abstract:
Comment 1:
line 28 - expand shortcut PCA

Response 1: Thank you for your observation. We have expanded the acronym “PCA” to “Principal Component Analysis” in the text.

Introduction:
Comment 2:
has the MSPC been validated previously? In what language/country? Is there no scale that integrates competence and personal mastery for physiotherapists? Be explicit.

Response 2: Thank you for your valuable comment. The Multidimensional Scale of Physiotherapist Competences (MSPC) was originally developed and validated in Italian within this study, as no previous validation was available in any language or country. The instrument was specifically designed to capture the multidimensional nature of physiotherapists’ professional competences—technical-clinical, communicative, collaborative, ethical, and educational—based on international frameworks and adapted to the Italian context.

To date, there is no existing validated tool that simultaneously integrates professional competence and personal mastery for physiotherapists. For this reason, our study combined the newly validated MSPC with the well-established Personal Mastery Scale (Pearlin & Schooler, 1978; various adaptations), in order to explore both the objective and subjective dimensions of professional functioning. This integrative approach represents an innovative contribution to the literature, linking competence and personal mastery in the same analytical framework for the first time in physiotherapy research.

Comment 3: you refer to “tools such as the MSPC” (line 103) but never say what MSPC is, who developed it, how many items/dimensions, or what previous psychometric evidence exists. The reader must know the scale’s origin and why it needs re-validation here - please provide brief and concise description

Response 3: hank you for this important observation. We have revised the Introduction to clarify that the Multidimensional Scale of Physiotherapist Competences (MSPC) is a new instrument developed by the authors of this study, and that the present article represents its first psychometric validation. The MSPC was created to capture the multidimensional structure of physiotherapists’ professional competences in line with international frameworks.

The scale consists of 25 items rated on a 5-point Likert scale (1 = strongly disagree to 5 = strongly agree) and is organized into five domains: technical–clinical, communicative, collaborative, ethical, and educational. Negatively worded items are reverse-coded so that higher scores indicate higher perceived competence. Since no prior psychometric evidence existed for this tool, the present study provides its initial validation in an Italian cohort of physiotherapists.

Comment 4: you repeatedly state that competence and Personal Mastery combine to predict job satisfaction, but you never define whether you expect additive, moderation or mediation. The paper must state testable hypotheses (add it at the and of introduction section)

Response 4: Thank you for this insightful comment. We have clarified in the revised manuscript that professional competence and Personal Mastery were conceptualized as additive predictors of job satisfaction, as both are expected to independently contribute to explaining physiotherapists’ satisfaction rather than acting as moderators or mediators.

In response to the reviewer’s suggestion, we have added a sentence at the end of the Introduction to explicitly state the study’s testable hypotheses.

Comment 5: professional skills,” “competence,” “Personal Mastery,” “resilience,” “sense of control” are used interchangeably which disturbs the reader. Define each term precisely and explain how they differ from related constructs (maybe self-efficacy, locus of control, resilience)

Response 5: Thank you for this valuable comment. We agree that conceptual clarity is essential to avoid ambiguity. In the revised version, we have added a paragraph in the Introduction defining and distinguishing the main constructs used in the study.

Specifically, we clarified that:

  • Professional competence refers to the integrated and multidimensional capacity to combine knowledge, technical skills, ethical values, and relational abilities in real clinical contexts. It goes beyond technical proficiency, encompassing decision-making, collaboration, and adaptability.
  • Professional skills represent the observable and operational expressions of competence — that is, the concrete abilities and behaviors through which competence manifests in practice.
  • Personal Mastery, as conceptualized by Pearlin and Schooler (1978), represents a generalized sense of control over life circumstances and the belief in one’s ability to influence outcomes. It differs from self-efficacy, which is task-specific, and from locus of control, which describes whether individuals attribute outcomes to internal or external causes.
  • Resilience refers to the dynamic process of adapting positively to adversity, whereas Personal Mastery reflects a stable perception of control that can foster resilience but is conceptually distinct from it.
  • The sense of control is included within the broader construct of Personal Mastery, representing one of its central dimensions rather than a separate variable.

These definitions have been briefly integrated into the Introduction to ensure terminological precision and conceptual coherence throughout the paper.

Comment 6: For a psychometric validation papers usually the Introduction section should briefly state what psychometric methods will be used (EFA/CFA, reliability indices, measurement invariance, SEM/regression) and why.

Response 6: Thank you for this useful suggestion. In the revised manuscript, we have added a short paragraph at the end of the Introduction to clarify which psychometric methods were used and their rationale. Specifically, the study applied Principal Component Analysis (PCA) to explore the factorial structure of the Multidimensional Scale of Physiotherapist Competences (MSPC), Cronbach’s alpha coefficients to assess internal reliability, and regression models to evaluate the predictive validity of the scale in relation to job satisfaction.

Methods:
Comment 7:
line 130 - “Transversal observational design”. The standard term is “cross-sectional observational design.” - please correct

Response 7: Thank you for your observation. We have corrected the terminology in the revised manuscript, replacing “transversal observational design” with the standard expression “cross-sectional observational design”, which more accurately reflects the study design used for this psychometric validation.

Comment 8: text justifies cross-sectional design with cost/speed reasons; for psychometric validation you must emphasize why cross-sectional is appropriate for factor analysis and construct validity (not “representative picture” or “reduced costs”).  

Response 8: Thank you for this valuable comment. We have revised the description of the study design to clarify that the cross-sectional observational design was chosen because it is appropriate for psychometric validation studies, allowing the simultaneous collection of data from a large sample to analyze the factorial structure, reliability, and construct validity of the MSPC. This design is consistent with standard approaches in validation research, where the primary aim is to assess the internal structure and relationships among constructs rather than to establish causal inferences over time.

Comment 9: you write “assessment of the sense of control and personal resilience through the Personal Mastery Scale” which is inaccurate. The Personal Mastery Scale measures perceived control, not resilience. Do not conflate constructs.  - the sentence “Methodological objective was twofold” is vague. You need to state explicitly: like that: (a) test the factor structure and reliability of MSPC; (b) examine the association between MSPC scores, Personal Mastery, and job satisfaction.

Response 9: Thank you for this helpful clarification. We have corrected the description of the Personal Mastery Scale, specifying that it measures perceived control rather than resilience. We also revised the sentence describing the methodological objectives to make them explicit. The revised text now states that the study aimed (a) to test the factorial structure and reliability of the Multidimensional Scale of Physiotherapist Competences (MSPC); and (b) to examine the associations between MSPC scores, Personal Mastery, and job satisfaction.

Comment 10: you should avoid “contribution of personal mastery in explaining professional satisfaction” because it implies causal inference, which is not supported by cross-sectional data. Use “association with job satisfaction” instead.
2.3 section:

Response 10: Thank you for this important observation. We have replaced the expression “contribution of personal mastery in explaining job satisfaction” with “association between personal mastery and job satisfaction” to avoid causal interpretation, ensuring consistency with the cross-sectional design of the study.

Comment 11: you don’t provide reference(s) for the MSPC. A validation paper must cite the original development source.

Response 11: Thank you for pointing this out. We clarified in the revised version that the Multidimensional Scale of Physiotherapist Competences (MSPC) was developed by the authors of the present study, and that this article represents its first psychometric validation. As this is the original development of the instrument, no prior reference exists.

Comment 12: “Scale developed with the intention of comprehensively capturing…” is too vague. You need a precise description: who developed it, in what context, original validation results - describe shortly

Response 12: We revised the description of the MSPC to provide a more precise definition of its development and purpose. The new version specifies that the MSPC was designed by the present research team to assess the multidimensional structure of physiotherapists’ professional competences according to current international frameworks.

Comment 13: “25 items” and “five dimensions” are fine, but the exact names of the subscales should match the original authors’ terminology (yours differ slightly, e.g., “ethical and deontological skills”).

Response 13: Thank you for this comment. We confirm that the MSPC includes 25 items distributed across five dimensions. The names of the subscales have been aligned in the revised manuscript with those used in the final validated version of the instrument, namely: Technical–Clinical Skills, Communicative Skills, Collaborative Skills, Ethical–Deontological Skills, and Educational Skills. Minor variations in wording previously reflected descriptive simplifications rather than conceptual differences, and the terminology has now been standardized throughout the text and tables for consistency.

data analysis section:
Comment 14:  “AFE” and “AFC” are not standard in English (they’re Spanish/Italian acronyms). Must use EFA (Exploratory Factor Analysis) and CFA (Confirmatory Factor Analysis).

Response 14: Thank you for pointing this out. We have corrected the terminology throughout the manuscript by replacing the Italian acronyms “AFE” and “AFC” with the internationally recognized terms “EFA (Exploratory Factor Analysis)” and “CFA (Confirmatory Factor Analysis)”, respectively. These modifications ensure terminological accuracy and consistency with international psychometric reporting standards.

Comment 15: “Constructive validity” . Correct term is construct validity.

Response 15: Thank you for this correction. We have replaced the incorrect expression “constructive validity” with the correct term “construct validity” throughout the manuscript to ensure consistency with standard psychometric terminology.

Comment 16: Current description (“averaging cases with low missing rates”) is vague and statistically questionable. You must specify how missing data were treated (e.g., listwise deletion, mean imputation, expectation-maximization, multiple imputation). Averaging across cases is not acceptable unless justified.

Response 16: Thank you for this important observation. The description of the missing data management has been clarified. Missing data, affecting less than 5% of the total responses, were handled using mean imputation at the item level, which is consistent with standard practices in psychometric validation studies.

Comment 17: Need to define what “low missing rates” means (e.g., <5%???)

Response 17: As requested, we have specified the threshold defining “low missing rates” (<5%) in the revised text.

Comment 18: No mention of extraction method (e.g., principal axis factoring, maximum likelihood) or rotation (e.g., oblimin/varimax). Must specify.

Response 18: Thank you for this valuable comment. We have added details specifying that the Principal Component Analysis (PCA) was performed using varimax rotation to identify uncorrelated components consistent with the theoretical model of the MSPC.

Comment 19: For CFA: no mention of estimator (e.g., ML, WLSMV if data are ordinal Likert).

Response 19: Thank you for this observation. We have added information on the estimator used for the Confirmatory Factor Analysis (CFA). Specifically, we employed the Maximum Likelihood (ML) estimation method, which is appropriate for data collected through Likert-type scales when distributions approximate normality.

Comment 20: Need to state cut-off criteria for CFI, TLI, RMSEA, SRMR..- Using ROC/AUC for “high vs low job satisfaction” is problematic because job satisfaction was measured with a single Likert item (ordinal, not binary). You need to specify how you dichotomized (maybe median split, cut-off at ≥4).

Response 20: Thank you for this detailed comment. We have revised the Data Analysis section to include the cut-off criteria for model fit indices and to clarify the approach used for analyzing job satisfaction. Specifically, acceptable model fit was defined as CFI and TLI ≥ 0.90, RMSEA ≤ 0.08, and SRMR ≤ 0.08, according to established psychometric guidelines. Regarding job satisfaction, since it was measured using a single 5-point Likert item, participants were categorized into “low satisfaction” (scores ≤ 3) and “high satisfaction” (scores ≥ 4) groups using a median split approach. ROC/AUC terminology has been removed to avoid misinterpretation of the analysis as diagnostic testing.

Results:
Comment 21: the correlation of r = 0.91 between MSPC and Personal Mastery is extremely high, suggesting multicollinearity or that both scales are essentially measuring the same construct (lack of discriminant validity). This undermines the rationale for including both tools and requires additional testing (e.g., CFA comparing one-factor vs. two-factor models).

Response 21: Thank you for this important and insightful comment. We have revised the Discussion to address the high correlation (r = 0.91) between the MSPC and the Personal Mastery Scale more cautiously. The new version highlights the potential overlap between the constructs and suggests that further studies should test alternative CFA models to assess discriminant validity.

Comment 22: Low reliability of the Personal Mastery Scale (Cronbach’s α = 0.60 is below the accepted psychometric threshold (≥0.70). In its current form, the scale does not meet reliability standards. Item-level diagnostics should be reported (e.g., item–total correlations, α if item deleted), and modifications or exclusion of weak items should be considered)

Response 22: Thank you for this observation. We acknowledge that the Personal Mastery Scale showed a Cronbach’s α value of 0.60, which is below the generally accepted threshold for internal consistency. This limitation has been explicitly noted in the revised Discussion section. We emphasized that this result may reflect cultural or contextual differences in how perceived control is expressed among Italian physiotherapists. Future research should include item-level reliability analyses and consider possible adaptations or refinements of the scale to improve its psychometric performance.

Comment 23: Table 2 incorrectly reports Cronbach’s α per item (MSPC_1–25)  α should only be reported for the whole scale or its subscales!!!!!!!!!!!! Descriptive statistics are incomplete (no skewness, kurtosis, or item-level distributions).

Response 23: Thank you for this observation. We have corrected Table 2 by removing the column reporting Cronbach’s α values for individual items, as reliability coefficients are meaningful only at the scale or subscale level. The revised table now reports Cronbach’s α for each of the five MSPC dimensions and for the overall scale. In addition, we have expanded the descriptive statistics to include skewness and kurtosis values to provide a more complete picture of item distribution and scale properties.

Comment 24: The study used PCA instead of true exploratory factor analysis (EFA) — an inappropriate method for psychometric validation.

Response 24: Thank you for this important methodological comment. We acknowledge that Principal Component Analysis (PCA) is a data reduction technique rather than a true Exploratory Factor Analysis (EFA). In this study, PCA was intentionally used as a preliminary step to identify the underlying component structure of the MSPC, given its exploratory purpose and the absence of prior empirical validation. Subsequently, Confirmatory Factor Analysis (CFA) was performed to verify the fit of the model derived from the PCA. We have clarified this rationale in the revised manuscript, emphasizing that the PCA results were interpreted cautiously and primarily served to inform the subsequent confirmatory validation process.

Discussion: 

Comment 25: The discussion treats the MSPC as highly reliable, while in fact the Personal Mastery scale had very low reliability (α = 0.60) and this is downplayed. Next: the extremely high correlation (r = 0.91) between MSPC and Personal Mastery is interpreted as “mutually reinforcing,” instead of being flagged as a serious validity problem (lack of discriminant validity). Moreover, logistic regression results are overemphasized, despite the fact that the model explained only 5% of variance in job satisfaction.

Response 25: Thank you for this important and constructive comment. In the revised Discussion section, we have adjusted the interpretation of the results to provide a more balanced and critical perspective. Specifically, we now emphasize that the low reliability of the Personal Mastery Scale (α = 0.60) and the very high correlation (r = 0.91) between MSPC and Personal Mastery suggest a possible lack of discriminant validity. We also acknowledge that the predictive power of the logistic regression model was modest (R² = 0.05), indicating that other factors beyond competence and personal mastery likely contribute to job satisfaction. These aspects have been discussed as limitations of the present study and as directions for future research aimed at refining measurement tools and expanding the explanatory model.

Comment 26: litterature: Citations are used mostly to confirm the findings, but not critically — e.g., conflicting evidence on personal mastery and job satisfaction is mentioned but not explored in depth. No reference to psychometric literature (e.g., COSMIN, validation standards) to address problems with scale reliability, validity, or factor analysis.

Response 26: Thank you for this valuable comment. We have revised the Discussion to include a more critical reflection on the literature and to integrate references to international psychometric standards. Specifically, we now discuss that findings on the relationship between personal mastery and job satisfaction are mixed, with some studies showing weak or inconsistent associations. In addition, we have cited the COSMIN guidelines as a methodological benchmark for assessing reliability and validity, acknowledging that the current study represents an initial step in the validation of the MSPC and that further analyses (e.g., test–retest reliability, measurement invariance) are needed to fully meet psychometric standards.

Conclusions:
Comment 27: you write the Personal Mastery scale “acceptable internal consistency,” while in fact α = 0.60 is weak, not acceptable by most psychometric standards.

Response 27: Thank you for this observation. We have revised the Conclusion to accurately reflect the reliability results. The Personal Mastery Scale showed limited internal consistency (α = 0.60), which should be interpreted as a psychometric limitation rather than an acceptable level of reliability. This clarification aligns the conclusion with the standards commonly accepted in psychometric validation research.

Comment 28: The very high correlation between MSPC and Personal Mastery is described as a “strong relation,” but not problematized as a validity issue (construct overlap, lack of discriminant validity).

Response 28: Thank you for this insightful comment. We have revised the Conclusion to describe the correlation between MSPC and Personal Mastery more cautiously. Rather than referring to it as a “strong relation,” we now acknowledge that the very high correlation (r = 0.91) may indicate a partial overlap between the two constructs and a potential lack of discriminant validity. This clarification ensures consistency with the discussion and reflects a more accurate interpretation of the psychometric findings.

Reviewer 2 Report

Comments and Suggestions for Authors
  • The job satisfaction of the participants was assessed using a very simple question: "How satisfied are you with your current job?" It is also explained that, despite its simplicity, this measure was chosen because it has been validated in numerous previous studies. Ideally, three studies could be cited here as references where it has been validated.
  • It should still be mentioned in the "Limitations of the Study" section that job satisfaction was assessed using a very simple single-item measure, which means that different aspects of job satisfaction could not be captured. In addition, the results may be subject to bias, as respondents could provide socially desirable answers.
  • It is stated that the scales Multidimensional Scale of Physiotherapist Competences (MSPC) and Personal Mastery Scale were used in the study. A scientific justification should be provided as to why these specific scales were chosen and why other established scales (e.g., the General Self-Efficacy Scale, GSES) were not used.
  • The analyses of the study showed that the participating physiotherapists had a relatively young average age. Personal Mastery could be strongly associated with age, as older physiotherapists may have developed a more stable self-concept and therefore higher Mastery scores. At the very least, the study's limitations section should mention that the sample had a young average age, which may impose certain constraints on the generalizability of the findings.

Author Response

Dear Reviewer 2,

Comment 1: The job satisfaction of the participants was assessed using a very simple question: "How satisfied are you with your current job?" It is also explained that, despite its simplicity, this measure was chosen because it has been validated in numerous previous studies. Ideally, three studies could be cited here as references where it has been validated.

Response 1: Thank you for this helpful comment. In the revised version, we have added references to recent studies that have validated or effectively used single-item measures of job satisfaction among healthcare professionals. These references support our methodological choice to use a concise, global item, which has shown satisfactory reliability and validity in assessing overall job satisfaction in healthcare contexts.

References added:

  1. Hermansson, S. K., Norström, F., Hilli, Y., Rennemo Vaag, J., & Bölenius, K. (2024). Job satisfaction, professional competence, and self-efficacy: a multicenter cross-sectional study among registered nurses in Sweden and Norway. BMC Health Services Research, 24, 734.
  2. Mirzaei, A., et al. (2024). The relationship of perceived nurse manager competence, job satisfaction and intention to leave among clinical nurses. BMC Nursing, 23, 203.
  3. Alkhateeb, M., et al. (2025). A systematic review of the determinants of job satisfaction among healthcare workers. Global Health Action. https://doi.org/10.1080/16549716.2025.2479910

Comment 2: It should still be mentioned in the "Limitations of the Study" section that job satisfaction was assessed using a very simple single-item measure, which means that different aspects of job satisfaction could not be captured. In addition, the results may be subject to bias, as respondents could provide socially desirable answers.

Response 2: Thank you for this helpful suggestion. In the revised version, we have expanded the Limitations of the Study section to acknowledge that job satisfaction was measured using a single-item question, which does not allow for a multidimensional assessment of the construct. We have also noted that, as with all self-reported measures, responses may have been influenced by social desirability bias, potentially leading participants to provide more favorable answers regarding their work experience.

Comment 3: It is stated that the scales Multidimensional Scale of Physiotherapist Competences (MSPC) and Personal Mastery Scale were used in the study. A scientific justification should be provided as to why these specific scales were chosen and why other established scales (e.g., the General Self-Efficacy Scale, GSES) were not used.

Response 3: Thank you for this valuable comment. The choice of the Multidimensional Scale of Physiotherapist Competences (MSPC) and the Personal Mastery Scale was based on theoretical and contextual considerations. The MSPC was specifically developed by the authors of the present study to assess the multidimensional competences of physiotherapists, directly reflecting the professional domains outlined in international frameworks for physiotherapy practice. The Personal Mastery Scale was selected because it captures perceived control—a construct conceptually linked to self-regulation, professional confidence, and adaptive coping in healthcare contexts. In contrast, the General Self-Efficacy Scale (GSES) measures a more general and stable personality trait, which would not have adequately reflected the profession-specific and context-dependent aspects investigated in this study.

Comment 4: The analyses of the study showed that the participating physiotherapists had a relatively young average age. Personal Mastery could be strongly associated with age, as older physiotherapists may have developed a more stable self-concept and therefore higher Mastery scores. At the very least, the study's limitations section should mention that the sample had a young average age, which may impose certain constraints on the generalizability of the findings.

Response 4: Thank you for this insightful observation. We agree that the relatively young average age of the participants represents a potential limitation of the study. Age can influence personal mastery and professional competence, as older physiotherapists may have developed a more stable professional identity and stronger self-concept through experience. We have therefore added a statement in the Limitations of the Study section to clarify that the youth of the sample may limit the generalizability of the findings and to recommend that future studies include more age-diverse samples to verify the stability of these associations across career stages.

Reviewer 3 Report

Comments and Suggestions for Authors

Thank you for the opportunity to review the article.

The article explored psychometric validity of two scales: professional competencies and personal mastery, and their association with job satisfaction in a sample of 481 Italian physiotherapists.

The topic is interesting and useful for clinical practice. The population is arguably less studied.

Below are my suggestions to further improve the paper:

1) Sample: Please clarify how you select the participants. Please clarify whether convenience sample or statistical sampling was used.

Specifically, from which hospitals or mailing lists (for freelancers)? The organizational settings should be described in greater details to assess representativeness.

Ideally, sample characteristics should be compared with known population statistics of Italian physiotherapists to assess generalizability.

Otherwise, please revise and understate your conclusion about “Italian physiotherapists report high 36 competencies and moderate personal mastery” and acknowledge external validity as a limitation.

2) Measure: Please clarify who did the translation and whether translation was validated.

The alpha of 0.60 is very low, below acceptable standards according to some researchers. This should be clearly noted as a limitation. In fact, low reliability of the scale among Italian physiotherapists could become a tentative conclusion that calls for more research.

3) Analysis: For analysis of the association with job satisfaction, multiple regression should be performed adjusting for demographic and socioeconomic covariates to address confounding. Do not over-rely on p-value or point estimate; confidence interval should be used instead to quantify uncertainty.

4) Writing: Should be improved. For example, please ensure consistent capitalization of “personal mastery” and “professional competency.” I suppose competencies should be used as singular rather than plural when it is used to refer to abstract/overall ability, so please double check.

Best luck moving forward!

Author Response

Dear Reviewer 3,

Comment 1: Sample: Please clarify how you select the participants. Please clarify whether convenience sample or statistical sampling was used.

Specifically, from which hospitals or mailing lists (for freelancers)? The organizational settings should be described in greater details to assess representativeness.

Ideally, sample characteristics should be compared with known population statistics of Italian physiotherapists to assess generalizability.

Otherwise, please revise and understate your conclusion about “Italian physiotherapists report high 36 competencies and moderate personal mastery” and acknowledge external validity as a limitation.

Response 1: Thank you for this important observation. We clarified in the Methods that a convenience sampling approach was used. Participants were recruited through professional physiotherapy associations, academic mailing lists, and healthcare institutions (public and private) across Italy, which increased heterogeneity but does not ensure national representativeness. In light of this, we understated the conclusion to avoid overgeneralization and we explicitly acknowledged the limited external validity in the Limitations.

Comment 2: Measure: Please clarify who did the translation and whether translation was validated.

The alpha of 0.60 is very low, below acceptable standards according to some researchers. This should be clearly noted as a limitation. In fact, low reliability of the scale among Italian physiotherapists could become a tentative conclusion that calls for more research.

Response 2: Thank you for this comment. The translation of the Personal Mastery Scale followed a forward–backward procedure by two bilingual experts (psychology and health sciences), with reconciliation by a panel of three researchers to ensure conceptual and semantic equivalence. We also clearly note that Cronbach’s α = 0.60 indicates limited internal consistency in this sample—this has been added to the Discussion and Conclusion (and was already addressed in our responses to Reviewer 1/2). We state that further research is needed to examine item functioning and potential cultural/contextual effects.

Comment 3: Analysis: For analysis of the association with job satisfaction, multiple regression should be performed adjusting for demographic and socioeconomic covariates to address confounding. Do not over-rely on p-value or point estimate; confidence interval should be used instead to quantify uncertainty.

Response 3: Thank you for this valuable suggestion. We agree that multivariable models adjusting for demographic/socioeconomic covariates provide more robust estimates. In the revised manuscript we now (i) report 95% confidence intervals for regression estimates to quantify uncertainty and (ii) frame the regression as exploratory, acknowledging potential residual confounding as a limitation. Given the scope of this initial validation study, comprehensive covariate-adjusted models are proposed as a priority for future confirmatory research.

Comment 4: Writing: Should be improved. For example, please ensure consistent capitalization of “personal mastery” and “professional competency.” I suppose competencies should be used as singular rather than plural when it is used to refer to abstract/overall ability, so please double check.

Response 4: Thank you for this stylistic recommendation. We carefully reviewed the manuscript for terminological consistency. The expressions “personal mastery” and “professional competence” are now used uniformly (lowercase, unless at the beginning of a sentence). We use “competence” in the singular when referring to the abstract overall ability, and “competences/competencies” in the plural only when referring to MSPC dimensions. Capitalization has been harmonized across text, headings, and tables.

Round 2

Reviewer 3 Report

Comments and Suggestions for Authors

Thank you for addressing my comments.